# Amphiphilic Polyethylene-*b*-poly(L-lysine) Block Copolymer: Synthesis, Self-Assembly, and Responsivity

**DOI:** 10.3390/ijms24065495

**Published:** 2023-03-13

**Authors:** Lixia Pei, Hongyu Ma, Yan Jiang, Handou Zheng, Haiyang Gao

**Affiliations:** 1School of Chemistry and Chemical Engineering, South China University of Technology, Guangzhou 510641, China; 2Daqing Chemical Engineering Research Center, Petrochemical Research Institute, Daqing 163714, China; 3School of Materials Science and Engineering, PCFM Lab, GD HPPC Lab, Sun Yat-sen University, Guangzhou 510275, China

**Keywords:** block copolymer, polyethylene, polypeptide, living/controlled polymerization, self-assembly

## Abstract

Polyethylene-*b*-polypeptide copolymers are biologically interesting, but studies of their synthesis and properties are very few. This paper reports synthesis and characterization of well-defined amphiphilic polyethylene-*block*-poly(L-lysine) (PE-*b*-PLL) block copolymers by combining nickel-catalyzed living ethylene polymerization with controlled ring-opening polymerization (ROP) of ε-benzyloxycarbonyl-L-lysine-*N*-carboxyanhydride (Z-Lys-NCA) and sequential post-functionalization. Amphiphilic PE-*b*-PLL block copolymers self-assembled into spherical micelles with a hydrophobic PE core in aqueous solution. The pH and ionic responsivities of PE-*b*-PLL polymeric micelles were investigated by means of fluorescence spectroscopy, dynamic light scattering, UV-circular dichroism, and transmission electron microscopy. The variation of pH values led to the conformational alteration of PLL from α-helix to coil, thereby changing the micelle dimensions.

## 1. Introduction

As one of the most important plastics and resins, polyethylene (PE) has been widely applied in daily life because of its excellent chemical and mechanical properties [1,2]. Furthermore, PE also has two striking features: high hydrophobicity and flexibility. PE is a highly hydrophobic polymer because its polymer chain has no polar groups or unsaturated bonds. Additionally, PE also exhibits high flexibility because it shows a very low glass transition temperature (*T*_g_ < −68 °C) [3]. Therefore, PE can be used as a featured polymer segment to construct block copolymer as functional PE-based polymeric materials [4,5].

Although a few functional PE block copolymers, such as polyethylene-*b*-polystyrene (PE-*b*-PS), polyethylene-*b*-poly(methyl methacrylate) (PE-*b*-PMMA), and polyethylene-*b*-poly(ε-caprolactone) (PE-*b*-PCL), have been successfully synthesized, their application is mainly limited to the field of polymer blends as a compatibilizer [6,7,8,9]. In fact, PE can be used as a good hydrophobic segment to construct amphiphilic copolymers with exceptionally rich phase behaviors, which broadens the potential applications of functional PE copolymers in the areas of biomaterials, sensors, and electronics [10,11,12,13,14,15,16].

Polypeptides as highly hydrophilic polymers show favorable biocompatibility and biodegradability [17,18,19]. Polypeptide conformational states in aqueous solutions have three different secondary structures: α-helix, β-sheet, and random coil, which are usually determined by a wide range of solution conditions, such as pH value, temperature, and salt concentration [20,21,22,23]. Two representative polypeptides are poly(L-glutamate) (PGA) with pendant carboxyl groups and poly(L-lysine) (PLL) with pendant amino groups. Because PE is structurally similar to lipid and polypeptide is structurally similar to protein, copolymers of ethylene and peptide are biologically interesting to use to study protein–lipid bilayer interactions, especially block copolymers [24]. However, the properties of very few polyethylene-*b*-polypeptide copolymers have been synthesized and studied because vastly different reactivities of two monomers make copolymer synthesis highly challenging [25,26]. Only structurally similar polyolefin-*b*-polypeptide copolymers, including polybutadiene-*b*-polypeptide [27,28,29,30,31,32], polyisoprene-*b*-polypeptide [33], and polypeptide-*b*-polyoctenamer-*b*-polypeptide copolymers [21], have been synthesized by a tandem synthetic strategy.

In this paper, we report the synthesis and characterization of new amphiphilic polyethylene-*block*-poly(L-lysine) (PE-*b*-PLL) diblock copolymers by a tandem synthetic strategy by combining nickel-catalyzed living ethylene polymerization with controlled ring-opening polymerization (ROP) of ε-benzyloxycarbonyl-L-lysine-*N*-carboxyanhydride (Z-Lys-NCA) and sequential post-functionalization. These block segments of PE-*b*-PLL copolymers have vastly different properties, such as solubility in solvents and secondary conformational states, which endow the amphiphilic block copolymer with rich phase behaviors. The self-assembly and pH and ionic responsivities of PE-*b*-PLL copolymers in aqueous solution were investigated for a better understanding of their potential applications.

## 2. Results and Discussion

### 2.1. Synthesis and Characterization of PE-b-PLL Block Copolymer

PE-*b*-PLL block copolymers were prepared by combining nickel-catalyzed living ethylene polymerization and ring-opening polymerization (ROP) of Z-Lys-NCA followed by deprotection of the benzyloxycarbonyl groups (Figure 1). First, PE block was prepared by living ethylene polymerization with amine–imine nickel catalyst developed by our group [34,35,36,37,38,39,40,41]. ZnEt_2_ as a chain transfer agent (CTA) was then charged into a catalytic system, and subsequent workup involving oxidation and hydrolysis reactions produced a hydroxyl-terminated polyethylene (PE–OH) with number-average molecular weight of 17 kg/mol (degree of polymerization (DP) = 607) and narrow polydispersity index (PDI) of ~1.02 (Figure 1) determined by gel permeation chromatography (GPC) [25,42,43,44]. PE–OH was converted to amino-terminated polyethylene (PE–COOCH(^i^Pr)NH_2_) by end-capping PE–OH with N-*tert*-butoxycarbonyl-L-valine (BOC-L-valine) and deprotecting BOC group [26,45]. The complete conversion from hydroxyl to amino group was evidenced by the full disappearance of the characteristic triplet peak of –C*H*_2_OH at 3.65 ppm and the complete disappearance of methyl signal of COOC(C*H*_3_)_3_ group at 1.46 ppm in the ^1^H NMR spectroscopies (Appendix A) [26].

Second, the PE macroinitiator (PE–COOCH(^i^Pr)NH_2_) was used to initiate the ROP of Z-Lys-NCA. Three polyethylene-*b*-poly(Z-Lys-NCA) (PE-*b*-PZL) copolymers were synthesized by adjusting the ratio of NCA monomer/macroinitiator. GPC curves of ROP polymerization products showed unimodal distributions (PDI = 1.3–1.4) (Figure 1), while copolymer molecular weights were not accurately determined because of the strong interactions between the polypeptides and GPC columns [26]. The block copolymers were further purified by precipitation from petroleum ether. The chemical structure of the block copolymer was proved by ^1^H NMR spectrum based on the characteristic peaks of PZL at 3.10, 4.42, 5.09, and 7.61 ppm (Figure 2). FT-IR analysis also supported the chemical structure of PE-*b*-PZL diblock copolymers (Figure 2). The characteristic bands of amide group at ~1650 and 1545 cm^−1^ were clearly observed in the FT-IR spectrum (Figure 3). Furthermore, the secondary conformation structures of PZL block were also proved by characteristic vibrational peaks [20,21]. In the vibrational range of the amide group, only two bands at 1650 (amide I) and 1550 cm^−1^ (amide II) were observed, strongly indicating that three block copolymer samples assumed α-helix conformation but no β-sheet conformation [25,46].

Third, amphiphilic PE-*b*-PLL block copolymers were prepared by hydrolysis reaction in the presence of acids (HBr/HAC) for deprotection of the benzyloxycarbonyl groups. PE-*b*-PLL diblock copolymers with pendant amino groups were easily soluble in water at weak acidic conditions (pH = 6.2), and pure PE-*b*-PLL copolymers extracted by water were used to study the structure characterization and properties in follow-up experiments. PE-*b*-PLL copolymers were characterized by ^1^H NMR spectrum in D_2_O. As shown in Figure 2, the characteristic resonances of PLL segment were observed, while no signals of PE block appeared because of the formation of the polymeric aggregates in water (see self-assembly below). In comparison with PE-*b*-PZL polymer precursor, PE-*b*-PLL copolymer did not show proton signals of amine (**h**), phenyl (**g**), or methylene (**f**) on the benzyloxycarbonyl group at 7.61, 7.29, or 5.09 ppm, strongly indicating full deprotection. As a result, amphiphilic PE-*b*-PLL copolymers with different PLL chain lengths were precisely prepared, and Table 1 summarizes their characterization results.

### 2.2. Self-Assembly of PE-b-PLL in Water

Amphiphilic PE-*b*-PLL copolymers with hydrophobic PE segments and hydrophilic PLL segments are expected to self-assemble spontaneously into polymeric aggregates in selective solvents. Three PE-*b*-PLL samples were directly dissolved in water at room temperature, and their aqueous solutions were used to study self-assembly.

The critical micelle concentrations (CMC) of three amphiphiles were first determined using pyrene as a fluorescent probe. Aqueous solutions of PE-*b*-PLL copolymers with different concentrations were prepared containing a constant pyrene probe concentration of 1 × 10^−6^ mol/L. The I_I_/I_III_ band intensity ratios of the pyrene emissions were plotted against the logarithm of amphiphile concentration [27]. As shown in Figure 4, PE_607_-*b*-PLL_275_ copolymer showed a clear deflection point at its CMC of 0.040 mg/mL (7.6 × 10^−7^ mol/L). It was also found that the CMC value decreased with decreasing length of PLL segment (Table 2) because of the low hydrophilic fraction in block copolymers. Generally, the three PE-*b*-PLL copolymers had very low CMC values (as low as 10^−7^ mol/L) because the PE segment has high hydrophobicity and PE-*b*-PLL copolymers readily form stable aggregates in aqueous solutions.

Dynamic light scattering (DLS) experiments measured at 25 °C and 0.20 mg/mL were further used to study the self-assembly properties of block copolymers in water. At pH = 6.2, PE-*b*-PLL copolymers formed aggregates with an average hydrodynamic radius (*R_h_*) in the range of 117 to 141 nm with unimodal size distributions (PDI < 0.15). It was observed that *R_h_* of PE-*b*-PLL polymeric aggregates decreased from 141 to 117 nm with a decrease in the length of PLL building block, which was a result of increasing hydrophilic PLL fraction. Transmission electron microscopy (TEM) was used to directly visualize the self-assembled aggregates of PE-*b*-PLL block copolymers. As shown in Figure 5, PE_607_-*b*-PLL_275_ copolymer in water self-assembled to spherical micelles with uniform size. The average radius of micelles determined by TEM was ~110 nm, which was consistent with the value measured by DLS analysis.

### 2.3. pH Responsivity of PE-b-PLL

As a kind of polyelectrolyte, PLL homopolymer has pH-responsive properties. Therefore, PE-*b*-PLL block copolymer is also expected to show pH responsivity. PE_607_-*b*-PLL_275_ was chosen as a representative sample to study pH responsivity. The effect of the pH value of the solution on the size of PE_607_-*b*-PLL_275_ was examined by DLS measurements. Figure 6A shows the pH-induced size changes on the hydrodynamic radius (*R_h_*) of PE_607_-*b*-PLL_275_ determined by DLS. Although a clear change of *R_h_* with alternation of solution from basicity to acidity was observed, the unimodal size distributions (~0.14) still remained (Figure 6B and Appendix A). The *R_h_* of PE_607_-*b*-PLL_275_ decreased with decreasing pH value from 9.0 (basicity) to 6.2 (near neutrality). When the pH value decreased from 6.2 to 1.5 (acidity), the *R_h_* of PE_607_-*b*-PLL_275_ also decreased. Further decreasing the pH value from 1.5 to 0.5 (strong acidity) did not change the *R_h_*. The biggest *R_h_* was observed at near neutral solution (pH ≈ 7). TEM images (Figure 5 and Figure 7) of polymeric micelles at different pH values also supported the pH-induced size change, and the biggest polymeric micelles were observed at pH of 6.2. These pH-induced changes of PE_607_-*b*-PLL_275_ polymeric micelle size are unique and different to previous studies [25,27,33].

Usually, the change of polymeric micelle size originates from the alteration in aggregation number. Furthermore, it is reported that PLL adopts different secondary conformational structures (α-helix, β-sheet, and coil) depending on the pH value of the solution, which can change micelle size [47,48,49]. Therefore, the effect of pH on secondary conformation of PLL block was investigated by UV-circular dichroism (CD) (Figure 8). As shown in CD spectra of PE_607_-*b*-PLL_275_ at a pH of 9.0, a characteristic inflected curve with small negative maxima at 209 and 220 nm was observed, proving an α-helical conformation. In contrast, a positive maximum at 218 nm and a negative minimum at 197 nm were observed in CD curves at acidic conditions (pH = 6.2 and 3.5), confirming a coil conformation [33]. The CD spectra determined at different pH values had an isodichroistic point at 206 nm. These observations clearly prove that the conformation transition of PLL block is an alternation from a coil conformation at acidic conditions to an α-helical conformation at basic conditions without the presence of β-sheet [25].

Based on DLS, TEM, and CD analyses mentioned above, the pH-induced changes of micelle size are reasonably explained. When the solution condition is changed from basicity to neutrality, the conformation alteration of PLL segment is responsible for a decrease in micelle size. When the solution condition is changed from neutrality to acidity, the conformation of the PLL segment is the same, and the decrease of micelle size is attributed to the aggregation number of the polymer chain. A schematic illustration of self-assembly of the PE-*b*-PLL into spherical micelles in aqueous solution is shown in Figure 2 [50,51,52]. In the polymeric micelles, hydrophobic PE segments are shrunken to form the micelle core in water, and hydrophilic PLL polymeric chains are located in the exterior of the micelles. PLL chains adopt coil conformation at acidic conditions and α-helical conformation at basic conditions.

### 2.4. Ionic Responsivity of PE-b-PLL

It is known that the addition of salt in the polypeptide solution often suppresses the pH responsivity because of the so-called “screening effect” of the charges on polypeptides [53]. Herein, NaCl was added into PE-*b*-PLL aqueous solutions to study the ionic effect on self-assembled polymeric micelles. As shown in Figure 6A (red line), the addition of NaCl into PE_607_-*b*-PLL_275_ solution led to a decrease in *R_h_* at any pH value. The effect of salt concentration was further studied at pH = 1.5 because of coil conformation of PLL segments and nearly invariable micelle sizes at pH = 1.5. As shown in Figure 9, *R_h_* markedly decreased with increasing NaCl concentration from 0 to 0.4 M, and then *R_h_* remained nearly invariable by further increasing NaCl concentration from 0.4 to 1.0 M. This observation strongly indicates that electrostatic interactions are fully screened at NaCl concentration of 0.4 M. Generally, the presence of NaCl minifies the pH responsivity because of a screening of the electrostatic character.

## 3. Materials and Methods

### 3.1. Synthesis of PE-b-PZL Block Copolymers

A round-bottom Schlenk flask with a stirring bar was heated for 3 h to 150 °C under vacuum and then cooled to room temperature. The 0.2 g PE–COOCH(^i^Pr)NH_2_ macroinitiator was dissolved in 5 mL of dried CHCl_3_ and then injected into the Schlenk reactor under N_2_. The desired amount of Z-Lys-NCA monomer solution in dried CHCl_3_ was charged into the reactor, and the reaction was continuously stirred for 4 days at 25 °C. The polymeric product was isolated when the solution was poured into an excess of petroleum ether. The resultant polymers were collected and purified by filtration, which involved washing with petroleum ether several times and drying in vacuum at 40 °C to a constant weight.

### 3.2. Synthesis of PE-b-PLL Block Copolymers

A total of 6 mL of HBr/HAc solution was added into 8 mL solution of PE-*b*-PZL block copolymers (0.5 g) in CH_3_Cl. After 1 h, hydrolysis reactions were stopped and poured into 200 mL ether. The precipitated polymers were collected and washed with diethyl ether. The crude polymers were further purified by extraction of water. The PE-*b*-PLL block copolymers were collected by removing water and drying in vacuum at 40 °C to a constant weight.

### 3.3. Preparation of the PE-b-PLL Polymeric Micelles

Amphiphilic PE-*b*-PLL solutions were prepared by direct dissolution in water at room temperature for 72 h. Aqueous solutions with various pH values and NaCl concentrations were prepared by dialysis of the solutions against water at various pH values and NaCl concentrations for 3 days. The aqueous solutions were allowed to equilibrate for at least 3 days under conditions of 25 °C before test.

### 3.4. Measurements

NMR spectra of polymers were carried out on a Bruker 500 MHz instrument (Bruker BioSpin, Billerica, Switzerland) in CDCl_3_ or D_2_O. Gel permeation chromatography (GPC) analysis of the molecular weight and PDI of the PE sample at 150 °C was performed on a high-temperature chromatography, PL-GPC 220 instrument (Agilent, CA, USA) equipped with a triple detection array. GPC analysis of the molecular weight and PDI of PE-*b*-PZL copolymer sample was performed on a Waters GPC system (Waters, Middleton, WI, USA) equipped with a refractive index detector at 40 °C. Fourier transform infrared (FT-IR) spectra were recorded on a Nicolet NEXUS-670 FTIR (Thermo Nicolet, Madison, WI USA) spectrometer. Fluorescence spectra were recorded on a VARIAN Cary Eclipse fluorescence spectrophotometer (Varian, California, USA) for CMC test. UV-circular dichroism (CD) analyses were performed at room temperature with a JASCO J 180 spectrometer (Jasco, Tokyo, Japan) employing quartz cells with 0.5 nm optical path length (185–250 nm). Dynamic light scattering (DLS) experiments were performed on a Malvern 300HSA Zetasizer instrument (Malvern Panalytical, Malvern, UK). Transmission electron microscopy (TEM) was performed on a TEM instrument (Philips TECNAI) (Philips-FEI, Eindhoven, The Netherlands) with an accelerating voltage of 120 kV. A negative staining technique was applied in observing self-assembly of PE-*b*-PLL amphiphiles.

## 4. Conclusions

In conclusion, we report the initial synthesis and characterization of well-defined amphiphilic PE-*b*-PLL block copolymers by combining nickel-catalyzed living ethylene polymerization with controlled ring-opening polymerization (ROP) of ε- Z-Lys-NCA and sequential post-functionalization. The prepared novel PE-*b*-PLL block copolymers are amphiphiles and can self-assemble into spherical micelles with a hydrophobic PE core in aqueous solution. The PLL segment endows PE-*b*-PLL block copolymer with pH and ionic responsivities in aqueous solution. The change of pH values leads to the conformational alteration of PLL from α-helix to coil, therefore changing micelle sizes. The presence of salt minifies the pH responsivity, and the electrostatic interactions are fully screened at NaCl concentration of 0.4 M. This kind of amphiphilic PE-*b*-PLL polymeric material shows potential for application in biomaterials.

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
