# Peer review of "Amphiphilic Polyethylene-b-poly(L-lysine) Block Copolymer: Synthesis, Self-Assembly, and Responsivity"

_ijms, 2023, doi:10.3390/ijms24065495_

Round 1

Reviewer 1 Report

The Authors reported the synthesis, self-assembly and pH responsivity of amphiphilic polyethylene-block-poly(L-lysine) copolymer. The ability to self-assembly into spherical micelles in water was confirmed by DLS and TEM and the responsivity to pH stimuli was also investigated.

In general, the study is interesting and the results and data interpretation in this manuscript are good but the authors need to address the following points before it can be accepted for publication.

In the introduction section, I suggest to increase the number of references regarding the different molecules that are able to self-assembly in water into micelles of spherical shape. Among others, I recommend to the authors to see and cite these papers:

-          ACS Omega 2018, 3, 3, 3143–3155. DOI:  10.1021/acsomega.7b01871

-          RSC Advances, 2020, 10, 9964. DOI: 10.1039/c9ra10981a

The Authors should be reported for all DLS results the peak value (with standard deviation) and the polydispersity index of the measurements (e.g in lines 149-150, 173-182).

I suggest to the Authors to add a low magnification TEM image, in order to provide to the readers an overview of the sample.

The most critical section of the manuscript is the Experimental Section. Authors need to better describe the experimental procedure. In particular expressions such as “appropriate” amount should be avoided (see main text at line 236 and in Supporting Synthesis of PE-OH). Please specify and indicate the quantity of the macroiniziator and what is the appropriate MMAO solution. Moreover, indicate the ration of HBr/HAc (line 244), specify the type of ether (line 246), indicate the amount of PE-b-PLL used for the preparation of the micelles (line 250) and ensure that all details and information of the experimental procedures are correctly reported to allow the experiment to be easily repeated. Finally, in line 256, only CDCl3 is indicated as NMR solvent, but D2O is also used. Please correct the sentence.  

Author Response

The Authors reported the synthesis, self-assembly and pH responsivity of amphiphilic polyethylene-block-poly(L-lysine) copolymer. The ability to self-assembly into spherical micelles in water was confirmed by DLS and TEM and the responsivity to pH stimuli was also investigated. In general, the study is interesting and the results and data interpretation in this manuscript are good but the authors need to address the following points before it can be accepted for publication.

(1) In the introduction section, I suggest to increase the number of references regarding the different molecules that are able to self-assembly in water into micelles of spherical shape. Among others, I recommend to the authors to see and cite these papers:

 ACS Omega 2018, 3, 3, 3143–3155. DOI:  10.1021/acsomega.7b01871

RSC Advances, 2020, 10, 9964. DOI: 10.1039/c9ra10981a

Response: We cite these two references. Besides, we also supply some recent references according to reviewer 3 suggestions. References are also re-numbered.

(2) The Authors should be reported for all DLS results the peak value (with standard deviation) and the polydispersity index of the measurements (e.g in lines 149-150, 173-182).

Response: DLS results mainly provide with the average hydrodynamic radius (Rh) of aggregates, therefore the peak value is not an important parameter. Besides, we provided figures of size distributions (B) of PE607-b-PLL275, and unimodal size distributions were observed in Figure 6B. According to reviewer suggestion, we supply the peak value and the polydispersity index of two typical samples in Figure 6B in Supplementary Materials. Besides, the standard deviation for Rh and R in Table 1 and 2, and error bars are also supplied in Figures. In the text, we also state that the polydispersity index is narrow and below 0.15.

(3) I suggest to the Authors to add a low magnification TEM image, in order to provide to the readers an overview of the sample.

Response: In Figure 5, we supply a low magnification TEM image to clearly observe an overview of the sample.

(4) The most critical section of the manuscript is the Experimental Section. Authors need to better describe the experimental procedure. In particular expressions such as “appropriate” amount should be avoided (see main text at line 236 and in Supporting Synthesis of PE-OH). Please specify and indicate the quantity of the macroiniziator and what is the appropriate MMAO solution. Moreover, indicate the ration of HBr/HAc (line 244), specify the type of ether (line 246), indicate the amount of PE-b-PLL used for the preparation of the micelles (line 250) and ensure that all details and information of the experimental procedures are correctly reported to allow the experiment to be easily repeated. Finally, in line 256, only CDCl3is indicated as NMR solvent, but D2O is also used. Please correct the sentence.  

Response: According to reviewer suggestion, we describe the experimental procedure in detail. The quantities of compounds are supplied in Experimental Section and Supplementary Materials. Besides, D2O is also added in NMR solvent. 

Reviewer 2 Report

The paper “Amphiphilic polyethylene-b-poly(L-lysine) block copolymer: Synthesis, self-assembly and responsivity” by the Lixia Pei and co-workers is devoted to preparation and characterization of new amphiphilic PE-b-PLL block copolymers. The object itself and its properties are undoubtedly of interest to a wide range of researchers. The authors have taken a conscientious approach to writing the article: all sections are clearly written and supported by adequate methods. Not being a deep specialist in this field, nevertheless, I can assert that the general chemical approach is well used and described. The article can be published in the current edition.

Author Response

Response: Thanks for your nice comments.

Reviewer 3 Report

This paper reports an investigation on synthesis, self-assembly and responsivity of amphiphilic polyethylene-b-poly(L-lysine) block copolymer:

In my opinion the paper can be accepted for publication in International Journal of Molecular Sciences after minor revisions, though Polymers seems to me a more suitable Journal for this paper.

 Polyethylene and polyolefins  have been synthesized and studied long before the paper cited in references cited as 1 and 2. Thus, please add more references.

In Table 1 and Table 2, please decrease the interline of the captions.

Page 5. In Figure 4 please change I1/I3 to II/ IIII. Moreover, I cannot read the insert in the figure.

Page 6. Line 178  Figures 5 and 7.

Page 8 Line 203 Figure 8 and not Figure 7

Author Response

This paper reports an investigation on synthesis, self-assembly and responsivity of amphiphilic polyethylene-b-poly(L-lysine) block copolymer: In my opinion the paper can be accepted for publication in International Journal of Molecular Sciences after minor revisions, though Polymers seems to me a more suitable Journal for this paper.

(1) Polyethylene and polyolefins have been synthesized and studied long before the paper cited in references cited as 1 and 2. Thus, please add more references.

Response: We supply some recent references including two literatures recommend by reviewer 1. Also, references are also re-numbered. Reference 1 is same to reference 38, which is deleted.

ACS Omega 2018, 3, 3, 3143–3155. DOI:  10.1021/acsomega.7b01871

 RSC Advances, 2020, 10, 9964. DOI: 10.1039/c9ra10981a

Polymers 2017, 9, 185.  Macromol. Chem. Phys. 2008, 209, 459–466

(2) In Table 1 and Table 2, please decrease the interline of the captions.

Response: We delete the column of PE molecular weight in Table 1 because of same values. Besides, we also format the Tables.

(3) Page 5. In Figure 4 please change I1/I3 to II/ IIII. Moreover, I cannot read the insert in the figure.

Response: I1/I3 is changed to II/ IIII in Figure 4.

(4) Page 6. Line 178  Figures 5 and 7.

Response: Figures 4 is corrected to Figure 5 in Line 178.

(5) Page 8 Line 203 Figure 8 and not Figure 7

Response: Figures 7 is corrected to Figure 8 in Line 203.
